# Anderson-Darling and Watson tests for the geometric distribution with estimated probability of success

**Héctor Francisco Coronel-Brizio**[1,2]☯*, **Alejandro Raúl Hernández-Montoya**[1,2]☯, **Manuel Enrique Rodríguez-Achach**[3]☯, **Horacio Tapia-McClung**[1]☯, **Juan Evangelista Trinidad-Segovia**[4]☯

**1** Instituto de Investigaciones en Inteligencia Artificial, Universidad Veracruzana, Xalapa, Veracruz, México, **2** Facultad de Física, Universidad Veracruzana, Xalapa, Veracruz, México, **3** Unidad Experimental Marista (UNEXMAR), Universidad Marista de Mérida, Mérida, Yucatán, México, **4** Departamento de Economía y Empresa, Universidad de Almería (UAL), Almería, España

☯ These authors contributed equally to this work.
* hcoronel@uv.mx

## Abstract

This paper introduces two new goodness-of-fit tests for the geometric distribution based on discrete adaptations of the Watson $W^2$ and Anderson-Darling $A^2$ statistics, where the probability of success is unknown. Although these tests are widely applied to continuous distributions, their application in discrete models has been relatively unexplored. Our study addresses this need by developing a robust statistical framework specifically for discrete distributions, particularly the geometric distribution. We provide extensive tables of asymptotic critical values for these tests and demonstrate their practical relevance through a financial case study. Specifically, we apply these tests to analyze price runs derived from daily time series of NASDAQ, DJIA, Nikkei 225, and the Mexican IPC indices, covering the period from January 1, 2015, to December 31, 2022. This work broadens the range of available tools for assessing goodness-of-fit in discrete models, which are essential for applications in finance and beyond. The Python programs developed for this paper are available to the academic community.

## 1 Introduction

A test of fit is a statistical methodology to assess how well a given theoretical distribution matches a sample of data. See Stephens *et al.* for a general discussion [1].

In particular, we are interested in the case of test of fits for the geometric distribution using the discrete Watson $W^2$ and Anderson-Darling $A^2$ statistics. The geometric distribution is used to model the number of Bernoulli trials occurring until observing the first success and is the discrete case of the exponential distribution. It has applications in several areas, such as hydrology to analyze water deficit for management of water, in Robotics for the location recognition of unmanned vehicles and reliability studies, in Medicine, to assess the risk of infection

**Data Availability Statement:** Data is publically available in Yahoo Finance website https://finance.yahoo.com/. Analyzed "runs" have been displayed

in the manuscript. and now in; https://doi.org/10.5281/zenodo.10659806.

**Funding:** Nacional de Investigadores, Consejo Nacional de Ciencia y Tecnología. México. Also, we thank financial support from projects with grants numbers 425854 and 5150 from the Consejo Nacional de Ciencia y Tecnología. CONACYT. https://conacyt.mx/, México. Dr. Juan E. Trinidad Segovia acknowledges the funding of the grant PID2021-127836NBI00 from the Ministerio Español de Ciencia e Innovación. The funders had no role in study design, data collection and analysis, decision to publish, or preparation of the paper.

**Competing interests:** The authors have declared that no competing interests exist.

with SARS-CoV-2 [2–4], in Ecology, to estimate the size of a population of individuals by means of a capture-recapture methodology [5, 6]. It has also been applied in the area of Statistical Process Control in order to ensure quality over time in a production or service, or to handle the multiple-reader interference problem [7, 8] among others, including finance to study the distribution of price "runs" (see [9] and references therein). Besides the well known textbook applications of the geometric distribution, the discrete case of the exponential distribution to phenomena like radioactive decay or quantum tunneling, we can mention the following few applications of this distribution in physical sciences: particle interactions in high energy Physics, photon detection in optics, physical chemistry, biochemistry and systems modeling, etc. [10–12].

## 1.1 Motivation

In our opinion, there is still work to be done in developing rigorous testing methods for assessing the goodness-of-fit for discrete models, particularly for the geometric distribution. While tests such as Anderson-Darling and Watson have been primarily developed for continuous distributions, relatively few alternatives have been adapted for discrete cases. This study aims to contribute to this issue by adapting the Anderson-Darling and Watson tests to a discrete models, in our case the geometric distribution and provide a valuable tool for researchers working with this data.

Our motivation for this work stems from a previous investigation into the geometric distribution's application to the duration of "price runs" in financial indices. In our earlier research [9], we analyzed the behavior of price runs in daily financial indices such as the NASDAQ, DJIA, and Nikkei 225. Through this study, we identified the need for a rigorous statistical test that could better assess the fit of geometric distributions in real-world financial data. The results of that research emphasized the limitations of current goodness-of-fit tests for discrete models, prompting the development of the current study.

## 1.2 Problem statement

Let us consider a geometric random variable $X$ with unknown parameter $\theta$, where $0 \leq \theta \leq 1.0$ is the probability of success on any given trial, and probability mass function:

$$p_i(\theta) = P(X = i|\theta) = (1 - \theta)^{i-1}\theta , \text{ for } i = 1, 2, \ldots \tag{1}$$

which gives the probability that the first success occurs on the *ith* trial.

The "Test of fit" problem consists in assessing whether a statistical model or theoretical distribution provides a good fit to the observed data. In the present work, given a random sample of $n$ values from Eq (1), statistical tests of fit using the discrete version of the well known Watson's $W^2$ and Anderson-Darling $A^2$ statistics are developed. This type of tests have already been constructed for some important discrete distributions such as the Poisson distribution (Spinelli and Stephens [13]), the first-digit Benford distribution (Lesperance *et al.* [14]) and for the discrete uniform distribution (Choulakian, Lockhart and Stephens, Lockhart, Spinelli and Stephens [15, 16]).

In the case that concerns us, interesting tests of fit for the geometric distribution also have been given, see for example [17], it based on a characterization of this distribution in relation to the conditional expectation of the second-order statistic, given the value of the first-order statistics. Also, the asymptotic null distribution of the estimated test statistic is estimated by means of the bootstrap technique. Reference [18] presents a geometric fit test based on the detection of gradual deviations from the geometric distribution and observed data, rather than

focusing on abrupt discrepancies. This statistical methodology is commonly referred to as "smooth test fit" and follows the work of Spinelli and Stephens [13].

[19] introduces and compares different goodness of fit tests for the Geometric distribution, including Anderson Darling and Cramér-von Mises tests. These tests are compared by simulation techniques. Finally, we must cite more theoretical and recent works that generalize the geometric distribution, which present several mathematical properties of this generalization, as well as methods employed to determine estimators for the new model based on maximum likelihood, moments, and proportion estimation [20].

By means of Monte Carlo simulations, reference [21] introduces and compares the relative performance of a few different statistical test for the geometric distribution mentioning that the best performed single statistic is the Anderson–Darling statistic.

The difference between the above mentioned kind of tests and the tests presented in this paper, is that for the geometric case and from asymptotic theory of test statistics, see section Asymptotic theory, we calculate the tests statistics asymptotic distributions for $W^2$ and $A^2$, and we give their respective Tables 1 and 2 for different values of the parameter $\theta$. We also provide an explicit explanation of the procedure necessary to apply the geometric fit tests developed here, see section Test procedure. We aim to test statistically the null hypothesis that the data sample to analyze was drawn from the geometric distribution. Finally, we are making the Python code needed to perform the corresponding calculations and tests available.

By way of example, we apply this methodology to daily "runs" constructed from daily time series of four financial market indices. These data were chosen because of the importance of these data sets in economic studies and also due to their broad and universal availability.

## 1.3 Preliminary definitions

Following [15], define $S_j = \sum_{i=1}^{j} o_i$, $T_j = \sum_{i=1}^{j} e_i$ and $Z_j = S_j - T_j$, where $o_i$ and $e_i = np_i(\theta)$ are the observed and expected number of observations in cells 1, 2, ..., respectively, with $p_i(\theta)$ given by Eq 1 and $n$ being the total number of observations in our dataset.

By definition $Z_j$ is the cumulative sum of the differences between the observed frequencies $o_i$, and the theoretically expected frequencies $e_i$ for all cells or categories from $i = 1$ up to $i = j$.

In this work we will consider the following definitions of the statistics:

$$W^2 = n^{-1} \sum_{i=1}^{\infty} Z_i^2 p_i(\theta) 0, \tag{2}$$

$$A^2 = n^{-1} \sum_{i=1}^{\infty} \frac{Z_i^2 p_i(\theta)}{H_i(1 - H_i)}, \tag{3}$$

where $H_i = \frac{T_i}{n}$ denotes the theoretical distribution function.

In practice, we will work with a finite number of cells. If we denote by $k$ the index corresponding to the last non-zero frequency cell, we define:

$$W^2(\theta) = n^{-1} \sum_{i=1}^{k} Z_i^2 p_i(\theta), \tag{4}$$

$$A^2(\theta) = n^{-1} \sum_{i=1}^{k} \frac{Z_i^2 p_i(\theta)}{H_i(1 - H_i)}. \tag{5}$$

If the parameter $\theta$ is unknown, it will be replaced by its Maximum Likelihood Estimator $\hat{\theta}$ to obtain the following expressions for the test statistics:

$$W^2(\hat{\theta}) = n^{-1} \sum_{i=1}^{k} Z_i^2 p_i(\hat{\theta}), \tag{6}$$

$$A^2(\hat{\theta}) = n^{-1} \sum_{i=1}^{k} \frac{Z_i^2 p_i(\hat{\theta})}{H_i(1 - H_i)}. \tag{7}$$

In our case, $\hat{\theta} = \bar{X}^{-1}$, where $\bar{X}$ denotes the arithmetic mean of the sample values.

## 2 Asymptotic theory

In this section, a brief summary of the distribution theory of the test statistics is given. For a more detailed description, the reader is referred, for example to [15].

### 2.1 Known $\theta$

Following [13], the test statistics given by Eqs (4) and (5) can be expressed as:

$$W^2(\theta) = \mathbf{Z}^T D \mathbf{Z}/n, \tag{8}$$

$$A^2(\theta) = \mathbf{Z}^T D G^{-1} \mathbf{Z}/n, \tag{9}$$

where $\mathbf{Z}$ is the vector with entries $Z_j$, for $j = 1, \ldots, k$; $D$ is a diagonal matrix whose $j$-th diagonal entry is $p_j(\theta)$ and $G$ is the diagonal matrix with elements $H_i(1 - H_i)$. The statistics have the general form:

$$Q_n = \mathbf{Z}^T V \mathbf{Z}/n, \tag{10}$$

where $V$ is a positive definite symmetric matrix. For $W^2(\theta)$, we have $V = D$ and $V = DG^{-1}$ for $A^2(\theta)$.

Let us denote by $\mathbf{o}$ and $\mathbf{e}$ the vectors of observed and expected values, respectively, of the cells; that is, $\mathbf{o}^T = [o_1, \ldots, o_k]$ and $\mathbf{e}^T = [e_1, \ldots, e_k]$.

Since $\mathbf{o}$ has a multinomial distribution with parameter $\mathbf{P}^T = [p_1(\theta), \ldots, p_k(\theta)]$, its mean vector and covariance matrix are $n\mathbf{P}$ and $n(D - \mathbf{PP}^T)$ and by the central limit theorem, $(\mathbf{o} - \mathbf{e})/\sqrt{n}$ converges to a multivariate normal distribution with mean vector zero and covariance matrix $\Sigma_0 = D - \mathbf{PP}^T$.

On the other hand, $\mathbf{Z} = R(\mathbf{o}\text{-}\mathbf{e})$, where $R$ denotes the lower-triangular matrix with unit elements (also called partial-sum matrix) and $\mathbf{Z}/\sqrt{n}$ converges to a multivariate normal distribution with mean vector $\mathbf{0}$ and covariance matrix $\Sigma = R\Sigma_0 R^T$ whose elements are given by min $(H_i, H_j) - H_i H_j$.

By defining $\mathbf{X} = V^{1/2}\mathbf{Z}/\sqrt{n}$, $\mathbf{X}$ converges to a normal distribution with mean vector $\mathbf{0}$ and covariance matrix $\Sigma_X = V^{1/2} \Sigma V^{1/2}$.

Also, $Q_n$ in Eq (10) can be expressed as:

$$Q_n = \mathbf{X}^T \mathbf{X} \xrightarrow{d} \sum_{i=1}^{k} \lambda_i v_i^2,$$

where $v_i, \ldots, v_k$ are independent standard normal random variables and $\lambda_1, \ldots, \lambda_k$ the eigenvalues of $\Sigma_X$.

## 2.2 Estimated $\theta$

When the parameter $\theta$ is unknown, it must me replaced by its maximum likelihood estimator (or other efficient estimator) $\hat{\theta}$. In this case, we define $\hat{\mathbf{e}}$ the vector with elements $\hat{e}_j = np_j(\hat{\theta})$, $j = 1, \ldots, k$. Under the usual regularity conditions [22], for the case of a single parameter, $(\mathbf{o} - \hat{\mathbf{e}})/\sqrt{n}$ converges to a normal random variable with mean vector $\mathbf{0}$ and covariance matrix $\hat{\Sigma}_0 = \Sigma_0 - \mathbf{b}\mathbf{b}^T/\mathbf{b}^T D^{-1}\mathbf{b}$, where the $j$-th element of $\mathbf{b}$ is:

$$b_j = \frac{dp_j(\theta)}{d\theta}, \quad j = 1, \ldots, k.$$

Again, the asymptotic distribution is that of:

$$\sum_{i=1}^{k} \lambda_i v_i^2,$$

but now, the eigenvalues are those corresponding to the matrix $\hat{\Sigma}_X = V^{1/2}\hat{\Sigma}V^{1/2}$ where $\hat{\Sigma} = R\hat{\Sigma}_0 R^T$.

## 2.3 Calculation of the asymptotic percentage points

In this work, the eigenvalues associated to each test statistic were obtained and the asymptotic percentage points were found using a Python implementation of the Lindsay-Pilla-Basak method [23–25]. According to the references, this method is accurate to two or three decimal places. For those interested, a R implementation of this method also exists [25, 26]. The percentage points were calculated for increasing values of $k$, until convergence was achieved. The final results are given in Tables 1 and 2.

In order to investigate the speed of convergence of the empirical percentage points, 25000 samples for selected values of $\theta$ and $n$ were simulated and the empirical percentage points for each statistic were computed. The results, shown in Tables 3 and 4, indicate that the asymptotic points can be used for moderately large values of $n$ (an usual case in practice) with good accuracy. In these tables the notation of infinity ($\infty$) refers to the theoretically calculated values based on the asymptotic distribution, which represents the distribution of the test statistics $W^2(\hat{\theta})$ and $A^2(\hat{\theta})$ when the sample size $n$ approaches infinity. These asymptotic values provide a theoretical benchmark derived from the limiting behavior of the statistics as the number of observations grows large.

## 3 Test procedure

In order to perform a test of fit for the geometric distribution (1), given $n$ observed values $x_1, \ldots, x_n$, the procedure is as follows:

1. Compute the sample estimate of the parameter $\theta$ as the inverse of the sample mean; i.e., $\hat{\theta} = \bar{x}^{-1}$.

2. Consider the first, say $k$, non empty cells and compute $p_i(\hat{\theta})$ using expression (1) for $i = 1, \ldots, k$, replacing $\theta$ with the value $\hat{\theta}$.

3. Compute the values of the statistics $W^2(\hat{\theta})$ and $A^2(\hat{\theta})$ using expressions (6) and (7) above.

4. Refer to Tables 1 and/or 2 corresponding to the test statistic, entering the table at the estimated value $\hat{\theta}$.

**Table 1. Asymptotic percentage points for statistic $W^2(\hat{\theta})$ for selected values of the parameter $\theta$ with values for different significance levels $\alpha$.**

**Upper tail probabilities for $W^2(\hat{\theta})$**

| $\theta$ | $\alpha = 0.50$ | $\alpha = 0.25$ | $\alpha = 0.15$ | $\alpha = 0.10$ | $\alpha = 0.05$ | $\alpha = 0.025$ | $\alpha = 0.01$ |
|---|---|---|---|---|---|---|---|
| 0.05 | 0.075 | 0.119 | 0.152 | 0.179 | 0.228 | 0.278 | 0.346 |
| 0.10 | 0.077 | 0.122 | 0.156 | 0.184 | 0.234 | 0.286 | 0.357 |
| 0.15 | 0.077 | 0.125 | 0.160 | 0.189 | 0.242 | 0.296 | 0.370 |
| 0.20 | 0.077 | 0.127 | 0.164 | 0.195 | 0.25 | 0.307 | 0.385 |
| 0.25 | 0.076 | 0.128 | 0.168 | 0.201 | 0.259 | 0.319 | 0.401 |
| 0.30 | 0.073 | 0.128 | 0.170 | 0.205 | 0.267 | 0.331 | 0.418 |
| 0.40 | 0.065 | 0.124 | 0.170 | 0.208 | 0.277 | 0.349 | 0.447 |
| 0.45 | 0.060 | 0.119 | 0.166 | 0.205 | 0.277 | 0.351 | 0.453 |
| 0.48 | 0.057 | 0.114 | 0.162 | 0.202 | 0.274 | 0.350 | 0.453 |
| 0.49 | 0.055 | 0.113 | 0.161 | 0.201 | 0.273 | 0.348 | 0.452 |
| 0.50 | 0.054 | 0.112 | 0.159 | 0.199 | 0.272 | 0.347 | 0.450 |
| 0.51 | 0.053 | 0.110 | 0.158 | 0.198 | 0.27 | 0.345 | 0.449 |
| 0.52 | 0.052 | 0.108 | 0.156 | 0.196 | 0.268 | 0.343 | 0.446 |
| 0.55 | 0.048 | 0.103 | 0.149 | 0.189 | 0.260 | 0.335 | 0.436 |
| 0.60 | 0.041 | 0.092 | 0.137 | 0.174 | 0.242 | 0.313 | 0.409 |
| 0.65 | 0.034 | 0.080 | 0.121 | 0.155 | 0.217 | 0.281 | 0.368 |
| 0.70 | 0.026 | 0.067 | 0.102 | 0.131 | 0.185 | 0.240 | 0.315 |
| 0.75 | 0.020 | 0.052 | 0.081 | 0.104 | 0.147 | 0.192 | 0.253 |
| 0.80 | 0.014 | 0.038 | 0.058 | 0.076 | 0.107 | 0.140 | 0.184 |
| 0.85 | 0.008 | 0.024 | 0.037 | 0.048 | 0.068 | 0.088 | 0.116 |
| 0.90 | 0.004 | 0.011 | 0.018 | 0.023 | 0.033 | 0.043 | 0.057 |
| 0.95 | 0.001 | 0.003 | 0.005 | 0.006 | 0.009 | 0.012 | 0.016 |

5. If the value of the test statistic exceeds the value for a given significance level $\alpha$, the null hypothesis that the sample was drawn from the distribution (1) is rejected for that level.

## 4 Example

First, let us remember that if the price of a security or the value of a financial index goes in the same direction, either increasing or decreasing, for $N$ consecutive days, we say that we have a bullish or bearish streak of $N$ consecutive days or a "run" of length $N$ days.

In reference [9], a data analysis of runs length is presented, in which we empirically compare the simple geometric statistical model, where $\theta = 1/2$ is the probability that the market goes up or down in a day, with daily data of financial indices. The same reference includes an extensive and up to date bibliography on classic and current runs research.

Time series of runs obtained for daily closing values from NASDAQ, DJIA, Nikkei 225 and the Mexican IPC stock indexes from January 1, 2015 to December 31, 2022 were analyzed, for each of these daily recorded time series, and the lengths in days of the corresponding uninterrupted daily trends (upward and downward) were calculated. Runs data were constructed from free financial data downloaded from the Yahoo finance website https://finance.yahoo.com/. As we mentioned before, our objective is to statistically test the null hypothesis suggesting that the sample was drawn from a geometric distribution.

Runs data for the four financial markets analyzed in this paper are shown in Table 5, where summary statistics and the calculated values of the test statistics are also given. a) Length refers

**Table 2. Asymptotic percentage points for statistic $A^2(\hat\theta)$ for selected values of the parameter $\theta$ with values for different significance levels $\alpha$.**

Upper tail probabilities for $A^2(\hat\theta)$

| $\theta$ | $\alpha = 0.50$ | $\alpha = 0.25$ | $\alpha = 0.15$ | $\alpha = 0.10$ | $\alpha = 0.05$ | $\alpha = 0.025$ | $\alpha = 0.01$ |
|---|---|---|---|---|---|---|---|
| 0.05 | 0.480 | 0.727 | 0.917 | 1.067 | 1.325 | 1.597 | 1.982 |
| 0.10 | 0.466 | 0.719 | 0.912 | 1.065 | 1.331 | 1.611 | 2.004 |
| 0.15 | 0.452 | 0.709 | 0.905 | 1.062 | 1.335 | 1.622 | 2.023 |
| 0.20 | 0.436 | 0.697 | 0.897 | 1.057 | 1.337 | 1.631 | 2.040 |
| 0.25 | 0.420 | 0.683 | 0.886 | 1.050 | 1.336 | 1.637 | 2.054 |
| 0.30 | 0.402 | 0.668 | 0.874 | 1.040 | 1.333 | 1.639 | 2.064 |
| 0.35 | 0.383 | 0.651 | 0.858 | 1.027 | 1.325 | 1.638 | 2.070 |
| 0.40 | 0.364 | 0.631 | 0.840 | 1.011 | 1.314 | 1.631 | 2.069 |
| 0.45 | 0.343 | 0.609 | 0.819 | 0.991 | 1.297 | 1.618 | 2.061 |
| 0.48 | 0.330 | 0.594 | 0.804 | 0.977 | 1.285 | 1.607 | 2.052 |
| 0.49 | 0.325 | 0.589 | 0.799 | 0.972 | 1.280 | 1.603 | 2.048 |
| 0.50 | 0.321 | 0.584 | 0.794 | 0.967 | 1.275 | 1.598 | 2.044 |
| 0.51 | 0.316 | 0.579 | 0.788 | 0.961 | 1.270 | 1.593 | 2.039 |
| 0.52 | 0.311 | 0.573 | 0.783 | 0.956 | 1.264 | 1.588 | 2.033 |
| 0.55 | 0.297 | 0.556 | 0.764 | 0.937 | 1.245 | 1.569 | 2.014 |
| 0.60 | 0.272 | 0.524 | 0.729 | 0.901 | 1.207 | 1.528 | 1.970 |
| 0.65 | 0.245 | 0.488 | 0.688 | 0.856 | 1.157 | 1.473 | 1.906 |
| 0.70 | 0.217 | 0.447 | 0.640 | 0.802 | 1.093 | 1.398 | 1.816 |
| 0.75 | 0.187 | 0.400 | 0.582 | 0.735 | 1.010 | 1.299 | 1.693 |
| 0.80 | 0.154 | 0.346 | 0.512 | 0.652 | 0.903 | 1.166 | 1.525 |
| 0.85 | 0.119 | 0.283 | 0.426 | 0.547 | 0.763 | 0.989 | 1.297 |
| 0.90 | 0.082 | 0.208 | 0.319 | 0.411 | 0.578 | 0.752 | 0.989 |
| 0.95 | 0.043 | 0.117 | 0.181 | 0.235 | 0.331 | 0.432 | 0.570 |

to the duration in days of each run, we observed uninterrupted trends lasting from one to thirteen days long. Number of observed runs for each market are shown, for example we observed for the NASDAQ 539 uninterrupted trends of one day length, 253 runs of two days length, etc.

b) $n$ indicates the total number of recorded runs. c) $\bar{x}$ denotes average length, d) $\hat\theta$ is $\bar{x}^1$, e) indicates the open 95% confidence intervals for the estimated $\theta$, and finally, g) and h) show the estimated values of $W^2$ and $A^2$ for the estimated $\theta$ respectively where $p$–values are included and denoted by $p_v$. Calculated values of the test statistics are discussed in next section Results.

## 5 Results

According to the estimated value of $\theta$, from Tables 1 and 2, it was found that in all cases, the $p$-values do not support the rejection of the null hypothesis. It is then concluded that the lengths of the trends can be appropriately described by the geometric distribution.

Also, from Table 5, it can be seen that the 95% confidence intervals [0.490, 0.534], [0.484, 0.528] and [0.487, 0.531], corresponding to the NASDAQ, DJIA and Nikkei 225 series, respectively, indicate that the null hypothesis $\theta = 0.5$ would not be rejected with reasonable significance levels, whereas for the case of the Mexican IPC series, the 95% confidence [0.454, 0.498] interval does not contain $\theta = 0.5$, which would produce an approximate $p$-value of 2.5% in testing the null hypothesis $\theta = 0.5$.

**Table 3. Empirical percentage points of the statistic $W^2(\hat{\theta})$ for selected values of the sample size $n$ and the parameter $\theta$ based on 25000 simulations and with different significance levels $\alpha$.**

Upper tail probabilities for $W^2(\hat{\theta})$

| $\theta$ | $n$ | $\alpha = 0.50$ | $\alpha = 0.25$ | $\alpha = 0.15$ | $\alpha = 0.10$ | $\alpha = 0.05$ | $\alpha = 0.025$ | $\alpha = 0.01$ |
|---|---|---|---|---|---|---|---|---|
| 0.10 | 25 | 0.078 | 0.122 | 0.156 | 0.183 | 0.229 | 0.278 | 0.346 |
| | 50 | 0.077 | 0.122 | 0.155 | 0.183 | 0.232 | 0.283 | 0.356 |
| | ∞ | 0.077 | 0.122 | 0.156 | 0.184 | 0.234 | 0.286 | 0.357 |
| 0.30 | 25 | 0.073 | 0.128 | 0.170 | 0.204 | 0.264 | 0.331 | 0.412 |
| | 50 | 0.074 | 0.128 | 0.170 | 0.206 | 0.264 | 0.330 | 0.425 |
| | ∞ | 0.073 | 0.128 | 0.170 | 0.205 | 0.267 | 0.331 | 0.418 |
| 0.50 | 25 | 0.054 | 0.110 | 0.159 | 0.194 | 0.262 | 0.336 | 0.433 |
| | 50 | 0.053 | 0.110 | 0.156 | 0.196 | 0.269 | 0.345 | 0.438 |
| | ∞ | 0.054 | 0.112 | 0.159 | 0.199 | 0.272 | 0.347 | 0.450 |
| 0.70 | 25 | 0.026 | 0.061 | 0.102 | 0.128 | 0.18 | 0.251 | 0.297 |
| | 50 | 0.025 | 0.067 | 0.100 | 0.123 | 0.178 | 0.234 | 0.307 |
| | 100 | 0.027 | 0.067 | 0.100 | 0.131 | 0.186 | 0.240 | 0.312 |
| | ∞ | 0.026 | 0.067 | 0.102 | 0.131 | 0.185 | 0.240 | 0.315 |
| 0.90 | 25 | 0.001 | 0.009 | 0.018 | 0.026 | 0.029 | 0.049 | 0.083 |
| | 50 | 0.004 | 0.008 | 0.014 | 0.023 | 0.035 | 0.048 | 0.072 |
| | 100 | 0.003 | 0.011 | 0.016 | 0.022 | 0.033 | 0.046 | 0.067 |
| | 200 | 0.004 | 0.011 | 0.017 | 0.023 | 0.033 | 0.043 | 0.060 |
| | ∞ | 0.004 | 0.011 | 0.018 | 0.023 | 0.033 | 0.043 | 0.057 |

## 6 Conclusion

Tests of fit for the geometric distribution based on the discrete version of the Watson $W^2$ and Anderson-Darling $A^2$ statistics are developed, particularly focusing on cases involving an unknown probability of success. Formulas to compute these two statistics when the parameter $\theta$ is unknown are provided in section Preliminary definitions by Eqs 6 and 7 respectively.

Additionally, we also present moderately extensive tables of asymptotic percentage points for each one of these two statistics and the procedure in the form of a list of instructions to fit data with a geometric distribution and assess the quality of this fit is provided in section Test procedure.

Beside, and as an illustration of the statistical methodology proposed for assessing a geometric test of fit, this research presents a comprehensive analysis of the geometric distribution's fit for financial series of daily uninterrupted trends or "runs", analyzed financial indices were Nasdaq, DJIA, Nikkei 25 and IPC.

An interesting finding of our study is that the trend lengths across the four major financial indices analyzed here predominantly adhere to a geometric distribution. This alignment is particularly notable with an estimated parameter value of $\theta$ close to 0.5, suggesting a near-equal probability of trend direction changes in these financial markets. However, our analysis also reveals variations in specific cases, such as the Mexican IPC series, where $\theta$ diverges from 0.5. This divergence is not just a statistical anomaly but potentially indicates a higher or lower likelihood of prolonged trends in this market.

Although signed trends were analyzed over a longer period than that used here, in reference [9], a similar divergence in the $\theta$ value and more prolonged trends are observed again in the case of the IPC market. This behavior could be a consequence of the fact that the IPC is an

**Table 4. Empirical percentage points of the statistic $A^2(\hat{\theta})$ for selected values of the sample size $n$ and the parameter $\theta$ based on 25000 simulations and with different significance levels $\alpha$.**

Upper tail probabilities for $A^2(\hat{\theta})$

| $\theta$ | $n$ | $\alpha = 0.50$ | $\alpha = 0.25$ | $\alpha = 0.15$ | $\alpha = 0.10$ | $\alpha = 0.05$ | $\alpha = 0.025$ | $\alpha = 0.01$ |
|---|---|---|---|---|---|---|---|---|
| 0.1 | 25 | 0.449 | 0.690 | 0.872 | 1.019 | 1.272 | 1.531 | 1.885 |
| | 50 | 0.460 | 0.708 | 0.895 | 1.046 | 1.313 | 1.574 | 1.951 |
| | 100 | 0.463 | 0.713 | 0.892 | 1.052 | 1.320 | 1.592 | 1.968 |
| | ∞ | 0.466 | 0.719 | 0.912 | 1.065 | 1.331 | 1.611 | 2.004 |
| 0.3 | 25 | 0.390 | 0.651 | 0.848 | 1.012 | 1.304 | 1.585 | 1.963 |
| | 50 | 0.396 | 0.657 | 0.859 | 1.027 | 1.323 | 1.614 | 2.031 |
| | 100 | 0.403 | 0.667 | 0.864 | 1.027 | 1.315 | 1.614 | 2.017 |
| | 200 | 0.406 | 0.665 | 0.863 | 1.031 | 1.322 | 1.639 | 2.091 |
| | ∞ | 0.402 | 0.668 | 0.874 | 1.040 | 1.333 | 1.639 | 2.064 |
| 0.5 | 25 | 0.315 | 0.567 | 0.777 | 0.937 | 1.243 | 1.571 | 1.961 |
| | 50 | 0.311 | 0.574 | 0.780 | 0.948 | 1.265 | 1.559 | 1.961 |
| | 100 | 0.316 | 0.581 | 0.779 | 0.958 | 1.275 | 1.605 | 2.009 |
| | 200 | 0.317 | 0.574 | 0.775 | 0.946 | 1.245 | 1.590 | 2.041 |
| | ∞ | 0.321 | 0.584 | 0.794 | 0.967 | 1.275 | 1.598 | 2.044 |
| 0.7 | 50 | 0.215 | 0.440 | 0.630 | 0.780 | 1.062 | 1.338 | 1.774 |
| | 100 | 0.209 | 0.440 | 0.628 | 0.784 | 1.073 | 1.390 | 1.753 |
| | 200 | 0.214 | 0.444 | 0.633 | 0.798 | 1.101 | 1.409 | 1.826 |
| | ∞ | 0.217 | 0.447 | 0.640 | 0.802 | 1.093 | 1.398 | 1.816 |
| 0.9 | 50 | 0.045 | 0.120 | 0.177 | 0.266 | 0.479 | 0.730 | 1.033 |
| | 100 | 0.061 | 0.154 | 0.220 | 0.293 | 0.475 | 0.705 | 0.959 |
| | 200 | 0.078 | 0.182 | 0.264 | 0.338 | 0.482 | 0.681 | 0.958 |
| | 500 | 0.081 | 0.210 | 0.316 | 0.404 | 0.562 | 0.731 | 0.935 |
| | ∞ | 0.082 | 0.208 | 0.319 | 0.411 | 0.578 | 0.752 | 0.989 |

emerging market, which is more immature and volatile compared to the other markets analyzed. Further studies are necessary to fully understand these results.

Such insights could be instrumental for investors and market analysts in understanding market dynamics and making informed decisions.

While we think that our findings are robust for the data sets and time frame considered, they are not without limitations. The assumption of a constant $\theta$ value over time might oversimplify the complex and dynamic nature of financial markets. Additionally, the methodology's reliance on historical data means it may not fully capture future market behaviors, especially in the face of unprecedented events or changes in market regulations.

Future research could expand upon this work in several ways. Firstly, exploring the variability of $\theta$ over different market conditions or time periods could provide a better understanding of market behavior. Secondly, applying these tests to a broader range of financial instruments, including emerging market indices and cryptocurrency markets, could validate the universality of the geometric distribution in financial trend analysis. Lastly, integrating machine learning techniques to predict changes in $\theta$ could offer groundbreaking tools for market prediction and investment strategy development.

In conclusion, our study contributes significantly to the statistical analysis of financial markets, offering a methodological framework that can be employed and expanded upon in various financial contexts. The insights gained highlight the intricate patterns underlying market trends and open avenues for future research to further understanding of these complexities.

**Table 5. Distribution of lengths of trends and summary statistics for all analyzed markets.** a) Observed runs duration; b) total number of runs; c) observed average run length; d) $\theta$ estimated; e) 95% C.I. for $\theta$; f) and g) estimated values of $W^2$ and $A^2$ statistics, respectively, with corresponding $p$–values.

| | Runs length (days) | No. Runs NASDAQ | No. Runs DJIA | No. Runs Nikkei | No. Runs IPC |
|---|---|---|---|---|---|
| a) | 1 | 539 | 518 | 502 | 449 |
| | 2 | 253 | 262 | 258 | 232 |
| | 3 | 121 | 116 | 118 | 131 |
| | 4 | 56 | 60 | 66 | 79 |
| | 5 | 36 | 30 | 19 | 40 |
| | 6 | 9 | 13 | 13 | 15 |
| | 7 | 8 | 12 | 7 | 5 |
| | 8 | 4 | 4 | 5 | 3 |
| | 9 | 2 | 2 | 2 | 1 |
| | 10 | 2 | 1 | 1 | 2 |
| | 11 | 2 | 0 | 0 | 0 |
| | 12 | 0 | 1 | 1 | 0 |
| | 13 | 0 | 0 | 1 | 1 |
| b) | $n$ | 1032 | 1019 | 993 | 958 |
| c) | $\bar{x}$ | 1.952 | 1.976 | 1.965 | 2.101 |
| d) | $\hat{\theta}$ | 0.512 | 0.506 | 0.509 | 0.476 |
| e) | 95% C.I. for $\theta$ | [0.490, 0.534] | [0.484, 0.528] | [0.487, 0.531] | [0.454, 0.498] |
| f) | $W^2(\hat{\theta})$ | 0.060 | 0.027 | 0.022 | 0.091 |
| | $p$–value ($p_v$) | $0.25 < p_v < 0.50$ | $p_v > 0.50$ | $p_v > 0.50$ | $p_v > 0.50$ |
| g) | $A^2(\hat{\theta})$ | 0.314 | 0.181 | 0.206 | 0.638 |
| | $p$–value ($p_v$) | $p_v > 0.50$ | $p_v > 0.50$ | $p_v > 0.50$ | $0.15 < p_v < 0.25$ |

## Acknowledgments

The authors want to thank MSc Selene Jimenez for her LaTeX typesetting help on the manuscript.

## Author Contributions

**Conceptualization:** Héctor Francisco Coronel-Brizio.

**Data curation:** Héctor Francisco Coronel-Brizio, Alejandro Raúl Hernández-Montoya, Horacio Tapia-McClung.

**Formal analysis:** Héctor Francisco Coronel-Brizio.

**Funding acquisition:** Alejandro Raúl Hernández-Montoya.

**Investigation:** Alejandro Raúl Hernández-Montoya, Manuel Enrique Rodríguez-Achach.

**Methodology:** Héctor Francisco Coronel-Brizio, Juan Evangelista Trinidad-Segovia.

**Resources:** Manuel Enrique Rodríguez-Achach, Horacio Tapia-McClung, Juan Evangelista Trinidad-Segovia.

**Software:** Manuel Enrique Rodríguez-Achach.

**Validation:** Horacio Tapia-McClung, Juan Evangelista Trinidad-Segovia.

**Visualization:** Juan Evangelista Trinidad-Segovia.

**Writing – original draft:** Héctor Francisco Coronel-Brizio.

**Writing – review & editing:** Alejandro Raúl Hernández-Montoya, Horacio Tapia-McClung, Juan Evangelista Trinidad-Segovia.

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
