## [Decision Letter · Decision Letter 0]

3 Oct 2023

PONE-D-23-22495Cramer-von Mises tests for the geometric distribution with estimated probability of successPLOS ONE

Dear Dr. Hernández-Montoya,

Thank you for submitting your manuscript to PLOS ONE. After careful consideration, we feel that it has merit but does not fully meet PLOS ONE’s publication criteria as it currently stands. Therefore, we invite you to submit a revised version of the manuscript that addresses the points raised during the review process based on the suggestions of the reviewers.

We look forward to receiving your revised manuscript.

Kind regards,

Oluwafemi Samson Balogun, Ph.D.

Academic Editor

PLOS ONE

Journal Requirements:

 "Authors acknowledge support from the Sistema Nacional de Investigadores, Consejo141

Nacional de Ciencia y Tecnología. México. Also we thank financial support from projects142

with grants numbers 425854 and 5150 from the Consejo Nacional de Ciencia y Tecnología.143

Conacyt. https://conacyt.mx/, México."

"Authors acknowledge support from the Sistema Nacional de Investigadores, Consejo

Nacional de Ciencia y Tecnolog´ıa. M´exico. Also we thank financial support from projects 

with grants numbers 425854 and 5150 from the Consejo Nacional de Ciencia y Tecnolog´ıa.

Conacyt. https://conacyt.mx/, M´exico."

"Authors acknowledge support from the Sistema Nacional de Investigadores, Consejo141

Nacional de Ciencia y Tecnología. México. Also we thank financial support from projects142

with grants numbers 425854 and 5150 from the Consejo Nacional de Ciencia y Tecnología.143

Conacyt. https://conacyt.mx/, México."

Reviewers' comments:

Reviewer's Responses to Questions

**Comments to the Author**

1. Is the manuscript technically sound, and do the data support the conclusions?

Reviewer #1: Yes

Reviewer #2: Yes

2. Has the statistical analysis been performed appropriately and rigorously? 

Reviewer #1: Yes

Reviewer #2: Yes

3. Have the authors made all data underlying the findings in their manuscript fully available?

Reviewer #1: Yes

Reviewer #2: Yes

4. Is the manuscript presented in an intelligible fashion and written in standard English?

Reviewer #1: Yes

Reviewer #2: Yes

5. Review Comments to the Author

Reviewer #1: This paper is clearly written and understandable, even to a physicist who is ot specialized in statistics.The applicability to financial markets is given, but the impact in the field is a matter of the future. Yet it seems very likely that other applications will arise and result important

Reviewer #2: I have reviewed the whole manuscript entitled “Cramer-von Mises tests for the geometric distribution with estimated probability of success”. However, I would like to make the following comments:

1. The paper needs to be proof-read for removal of spelling and grammatical errors.

2. There are some obvious improvements in the introductory and literature review parts of the manuscript.

3. Problem Statement needs to be improved.

4. Interpretation of the Table or data analysis part should be elaborated.

5. Conclusion of the study should be elaborated, currently, it is merely a summary of the research. It is very important to pinpoint the implications of your study.

6. Delete the low-level journal references and non-scientific references from the paper.

6. PLOS authors have the option to publish the peer review history of their article (what does this mean?). If published, this will include your full peer review and any attached files.

Reviewer #1: No

Reviewer #2: **Yes: **Dr Chinnadurai Kathiravan

---

## [Author Response · Author response to Decision Letter 0]

18 Feb 2024

Dear Professor Oluwafemi Samson Balogun, Ph. D.

Academic Editor

PLOS ONE

We extend our sincere appreciation for your dedicated time and effort as academic editor for our article, as well as for the diligent work of the referees who reviewed our paper.

Below, we summarize the changes and corrections made to our manuscript, along with our responses to the referees' feedback:

1) Our manuscript now adheres to PLOS ONE's style requirements, including file naming conventions.

2) The code and data used to construct analyzed runs is now available at the following website: https://doi.org/10.5281/zenodo.10659806

3) We have clarified the role of funders as follows: "The funders had no role in study design, data collection and analysis, decision to publish, or preparation of the manuscript."

4) Funding information has been removed from the Acknowledgments section of our manuscript.

5) Responses to reviewers' comments and suggestions:

Starting with referee number 1, we appreciate the positive feedback provided. In response, we have added a brief introduction to our paper highlighting various applications of the geometric distribution in hydrology, technology, high energy physics, etc., supported by relevant citations.

Regarding referee #2's comments:

“1. The paper needs to be proof-read for removal of spelling and grammatical errors.”

We have thoroughly proofread the entire manuscript to eliminate spelling and grammatical errors. 

“2. There are some obvious improvements in the introductory and literature review parts of the manuscript.”

We have written a genuine introduction and reorganized this by adding two subsections as will be explained latter. Please refer to lines 2 to 56 of the revised paper.

With regard to the bibliography, it has been corrected and expanded; in addition, the articles that were not referenced in the text of the first version of the article have been removed. 

For more details on it, see point “6)” and sub-items b), c), and d) below.

“3. Problem Statement needs to be improved.”

The Problem Statement has been enhanced and clarified and section introduction has been reorganized adding a subsection titled "1.1 Problem Statement". Please refer to lines 20 to 52 of the revised paper.

“4. Interpretation of the Table or data analysis part should be elaborated.”

We have elaborated on the interpretation of the table and data analysis by explicitly explaining all table variables within the paper text and table labels. Please, see sub-items g) and h) below and lines 143 to 152 of the new version of our paper and explanatory label of table number 5.

“5. Conclusion of the study should be elaborated, currently, it is merely a summary of the research. It is very important to pinpoint the implications of your study.”

The Conclusion section has been revised and rewritten to emphasize the implications of our findings, their applications, the limitations of our study and potential future research directions. See lines 165 to 209 of our paper.

“6. Delete the low-level journal references and non-scientific references from the paper.”

We have updated the reference list and highlighted all changes in the Revised Manuscript. We have also addressed concerns about low-level and non-scientific references, but we kindly request specific guidance on which references should be removed if necessary. We assume that reviewer #2 refers to those relate to the python programs used, where we do give credit to those whose work helped us to perform the presented calculations.

Below, we detail a summary of changes and corrections made.

a) The title of the article has been changed to provide a closer explanation of the topic being discussed.

The original title was "Cramer-von Mises tests for the geometric distribution with estimated probability of success," and the new title will be: "Anderson-Darling and Watson tests for the geometric distribution with estimated probability of success."

b) Section “1 Introduction” has been rewritten, indicating importance and applications of Geometric distributions with respective references. See lines 1 to 19. 

c) Still in “Section 1 Introduction”, subsection “1,1 Problem statement” has been introduced. See lines 21 to 52.

d) Again in the introduction, a new section titles “1,2 Preliminary definitions” has been added to clarify text. Minor corrections to original text here included in lines 54 to 56

e) Although the labeling of the tables was correct, their presentation order was wrong, so this have been corrected in addition to that respective labels have been enhanced with a clearer explanation of them, specifically:

Labels of tables 1 and 2:

Addition of the phrase “...for different significance levels α” at the end of respective labels.

f) Labels of tables 3 and 4:

Addition of the phrase “...with different significance levels α” at the end of corresponding labels.

g) Label of table 5:

Before:

Table 3. Distribution of lengths of trends from series of returns and summary statistics.

Now:

Table 5. Distribution of lengths of trends and summary statistics for all analyzed markets. 

a) Observed runs duration; b) total number of runs; c) observed average run length; d) θ estimated; e) 95% C.I. for θ; f) and g) estimated values of W 2 and A2 statistics, respectively, with corresponding p−values.

Also, table 5 has now an additional column, the first, to make a better explanation of it.

h) Explanation of table 5 has been added in text. See lines 143 to 152.

i) As required by referee #2, section "Conclusion" has been revised and rewritten, mentioning the implications of our findings, their applications, the limitations of our study, and potential future research directions. See lines numbers 165 to 209.

Finally, many minor corrections have been made, which can be seen in the underlined version of our manuscript.

Dear Professor Oluwafemi, once again, we express our gratitude to you and our two reviewers for your valuable feedback and guidance throughout the review process.

---

## [Decision Letter · Decision Letter 1]

22 Mar 2024

PONE-D-23-22495R1Anderson-Darling and Watson tests for the geometric distribution with estimated probability of success.PLOS ONE

Dear Dr. Alejandro Raul Hernandez-Montoya,

Thank you for submitting your manuscript to PLOS ONE. After careful consideration, we feel that your manuscript will likely be suitable for publication if it is revised to address the points below.   Therefore, my decision is "Minor Revision".

We invite you to submit a revised version of the manuscript that addresses the points raised during the review process.

Please revise this paper.

We encourage you to submit your revision by May 06 2024 11:59PM. If you will need more time than this to complete your revisions, please reply to this message or contact the journal office at plosone@plos.org. Please include the following items when submitting your revised manuscript:

We look forward to receiving your revised manuscript.

Kind regards,

Oluwafemi Samson Balogun, Ph.D.

Academic Editor

PLOS ONE

Journal Requirements:

Reviewers' comments:

Reviewer's Responses to Questions

**Comments to the Author**

1. If the authors have adequately addressed your comments raised in a previous round of review and you feel that this manuscript is now acceptable for publication, you may indicate that here to bypass the “Comments to the Author” section, enter your conflict of interest statement in the “Confidential to Editor” section, and submit your "Accept" recommendation.

Reviewer #3: All comments have been addressed

Reviewer #4: All comments have been addressed

2. Is the manuscript technically sound, and do the data support the conclusions?

Reviewer #3: Yes

Reviewer #4: Yes

3. Has the statistical analysis been performed appropriately and rigorously? 

Reviewer #3: Yes

Reviewer #4: Yes

4. Have the authors made all data underlying the findings in their manuscript fully available?

Reviewer #3: Yes

Reviewer #4: Yes

5. Is the manuscript presented in an intelligible fashion and written in standard English?

Reviewer #3: Yes

Reviewer #4: Yes

6. Review Comments to the Author

Reviewer #3: Some points need to be addressed in the current version of the manuscript. Please see the attachment.

Reviewer #4: (No Response)

7. PLOS authors have the option to publish the peer review history of their article (what does this mean?). If published, this will include your full peer review and any attached files.

Reviewer #3: No

Reviewer #4: No

---

## [Author Response · Author response to Decision Letter 1]

6 Jun 2024

Dear Professor Oluwafemi Samson Balogun, Ph. D.

Academic Editor

PLOS ONE

Here are our responses to the comments and suggestions from referee number 4. Once again, we appreciate the time you and our referees have dedicated to evaluating our article.

1. In Equation (1), a range of θ must be specified.

The range 0 ≤ θ ≤ 1 has been specified, and the text “which gives the probability that the first success occurs on the ith trial.” has been included. See lines 21 and 23 respectively of revised manuscript.

2. In Equations (2)-(7), what is Zi ?

We have included the following text in lines 60 to 62 of the revised paper:

By definition Zj is the cumulative sum of the differences between the observed frequencies o_i , and the theoretically expected frequencies e_i for all cells or categories from i =1 up to i = j.

3. When the momentchi2 is employed [1], how to implement it?

We utilized the library by calling its function Momentchi2, using the lpb4 functionality (Lindsay-Pilla-Basak method) within the context of the proposed analysis according to the official documentation, ensuring that necessary parameters were correctly passed and following recommended practices to be sure the routine runs correctly. We used the lpb4 option in the momentchi2 library because it offers a highly efficient and accurate method for computing the cumulative distribution function (CDF) for chi-squared distributions. This option is particularly suitable for our analysis as it establishes precise results with lower computational cost, which is crucial for handling large datasets and complex statistical models, especially when computer equipment is not last generation, such as is our case.

4. It seems R is used in this study, it may be cited [2].

We did not use R in our study, instead we have used python.

5. R source code may be provided.

We have provided python code. See https://zenodo.org/doi/10.5281/zenodo.10519808

Given its relevance to academics and others interested in these topics, we have cited reference [2] related to the R package and mentioned by referee No. 4 (cited as [24] in paper), reference [1] mentioned by our referee it is already mentioned since previous revised version of our paper (reference [22] in paper). Those references are:

[1] D. Bodenham. momentchi2: Moment-Matching Methods for Weighted Sums of Chi-Squared

Random Variables, 2016. R package version 0.1.5.

[2] R Core Team. R: A Language and Environment for Statistical Computing. R Foundation for

Statistical Computing, Vienna, Austria, 2023.

Dear Professor Oluwafemi, once again, we express our gratitude to you and all our reviewers for valuable feedbacks and guidance throughout the review process. 

Warm regards, 

On behal of of all authors, 

Dr. Alejandro Raúl Hernández Montoya 

Universidad Veracruzana, 

alhernandez@uv.mx

www.uv.mx/personal/alhernandez

---

## [Decision Letter · Decision Letter 2]

4 Oct 2024

PONE-D-23-22495R2

Anderson-Darling and Watson tests for the geometric distribution with estimated probability of success.

PLOS ONE

Dear Dr. Hernández-Montoya,

Thank you for submitting your manuscript to PLOS ONE. After careful consideration, we feel that it has merit but does not fully meet PLOS ONE’s publication criteria as it currently stands. Therefore, we invite you to submit a revised version of the manuscript that addresses the points raised during the review process.

We look forward to receiving your revised manuscript.

Kind regards,

Ahmed Mahmoud Gemeay

Academic Editor

PLOS ONE

Additional Editor Comments:

I find this approach interesting, but more details on the new model should be added using the following significant comments.

1- What is the motivation and contribution of this paper? Authors must write two or three paragraphs in the introduction to show the novelty and importance of their paper.

2 - Authors should add more recent papers discussing related problems and their relation to their paper if they exist.

3 - Rewrite the abstract again to be more attractive.

4 - I need clarification as to why these two measures are new for discrete models; the authors should have explained them in the paper.

5 - Authors used MLE to find their estimator; can any other estimation method be used? Can they add more studies on this estimator, such as unbiased property?

6 - What symbol does infinity refer to in Tables 1 and 2? What size do authors use to write this symbol?

7 - Can authors add another example from another field?

8 - The English language should be revised.

9 - All equations must be revised to see if they need a comma, a point, or a number at the end of them.

Reviewers' comments:

Reviewer's Responses to Questions

**Comments to the Author**

1. If the authors have adequately addressed your comments raised in a previous round of review and you feel that this manuscript is now acceptable for publication, you may indicate that here to bypass the “Comments to the Author” section, enter your conflict of interest statement in the “Confidential to Editor” section, and submit your "Accept" recommendation.

Reviewer #4: All comments have been addressed

2. Is the manuscript technically sound, and do the data support the conclusions?

Reviewer #4: Yes

3. Has the statistical analysis been performed appropriately and rigorously? 

Reviewer #4: Yes

4. Have the authors made all data underlying the findings in their manuscript fully available?

Reviewer #4: Yes

5. Is the manuscript presented in an intelligible fashion and written in standard English?

Reviewer #4: Yes

6. Review Comments to the Author

Reviewer #4: The authors addressed all the issues raised in previous review. The explaination presented by authors is justified.

7. PLOS authors have the option to publish the peer review history of their article (what does this mean?). If published, this will include your full peer review and any attached files.

Reviewer #4: No

---

## [Author Response · Author response to Decision Letter 2]

25 Oct 2024

Xalapa, Veracruz. México. October/21/2024.

Dear Professor Ahmed Mahmoud Gemeay

Academic Editor

PLOS One

We have carefully addressed each of your comments and suggestions regarding our paper, and our detailed responses are provided below. We truly appreciate the time and effort you have dedicated to reviewing our manuscript, and we hope that our responses are clear and satisfactory.

I find this approach interesting, but more details on the new model should be added using the following significant comments.

1- What is the motivation and contribution of this paper? Authors must write two or three paragraphs in the introduction to show the novelty and importance of their paper.

 We explained our motivation and contribution on the paper. To explain the motivation that led this paper we have included the following text in a subsection named “Motivation” in the introduction section: 

“subsection{Motivation}

In our opinion, there is still work to be done in developing rigorous testing methods for assessing the goodness-of-fit for discrete models, particularly for the geometric distribution. While tests such as Anderson-Darling and Watson have been primarily developed for continuous distributions, relatively few alternatives have been adapted for discrete cases. This study aims to contribute to this issue by adapting the Anderson-Darling and Watson tests to a discrete model, in our case the geometric distribution and provide a valuable tool for researchers working with this data.”

“Our motivation for this work stems from a previous investigation into the geometric distribution’s application to the duration of “price runs” in financial indices. In our earlier research \\cite{Hernandez}, we analyzed the behavior of price runs in daily financial indices such as the NASDAQ, DJIA, and Nikkei 225. Through this study, we identified the need for a rigorous statistical test that could better assess the fit of geometric distributions in real-world financial data. The results of that research emphasized the limitations of current goodness-of-fit tests for discrete models, prompting the development of the current study.”

We have already described our contribution in the text or our paper: 

“The difference between the above mentioned kind of tests and the tests presented in this paper, is that for the geometric case and from asymptotic theory of test statistics, see section \\ref{sec:AsymtoticTheory}, we calculate the tests statistics asymptotic distributions for $W^{2}$ and $A^{2}$, and we give their respective Tables \\ref{tableW2} and \\ref{tableA2} for different values of the parameter $\\theta$. We also provide an explicit explanation of the procedure necessary to apply the geometric fit tests developed here, see section \\ref{sec:TestProcedure}. We aim to test statistically the null hypothesis that the data sample to analyze was drawn from the geometric distribution. Finally, we are making the Python code needed to perform the corresponding calculations and tests available.”

2 - Authors should add more recent papers discussing related problems and their relation to their paper if they exist.

Although we were able to find many papers on the geometric distribution, there are not many recent ones that discuss fitting it and describing a goodness-of-fit test. Instead, we found numerous papers on generalizing the geometric distribution, and one on generalizing and fitting it, which was the closest and recent reference we could cite, and its reference number is 20.

We also cite a work, although not recent, that serves as a precursor to ours. It presents goodness-of-fit tests for the geometric distribution based on discrete adaptations of the Anderson-Darling A^{2} test and others, and compares them with each other. Its reference number is 19.

3 - Rewrite the abstract again to be more attractive. 

We appreciate your suggestion; the abstract has been modified in order to make it more attractive:

New abstract:

“We propose two novel goodness-of-fit tests for the geometric distribution based on discrete adaptations of the Watson $W^{2}$ and Anderson-Darling $A^{2}$ statistics, where the probability of success is unknown. Although these tests are widely applied to continuous distributions, their application in discrete models has been relatively unexplored. Our study addresses this need by developing a robust statistical framework specifically for discrete distributions, particularly the geometric distribution. We provide extensive tables of asymptotic critical values for these tests and demonstrate their practical relevance through a financial case study. Specifically, we apply these tests to analyze price runs derived from daily time series of NASDAQ, DJIA, Nikkei 225, and the Mexican IPC indices, covering the period from January 1, 2015, to December 31, 2022. This work broadens the range of available tools for assessing goodness-of-fit in discrete models, which are essential for applications in finance and beyond. The Python programs developed for this paper are available to the academic community.”

4 - I need clarification as to why these two measures are new for discrete models; the authors should have explained them in the paper.

The measures are not new as we state in the text:

“This type of tests has already been constructed for some important discrete

distributions such as the Poisson distribution ( Spinelli and Stephens [1] ), the firts-digit Benford distribution ( Lesperance et al. [3]) and for the discrete uniform distribution ( Choulakian, Lockhart and Stephens [2].)”

Our new contribution is the methodology presented, that lies in adapting these tests for use with the geometric model. The tests we present offer a robust approach for discrete models. we have calculated detailed asymptotic tables for these tests, which extend what has been previously available, particularly for discrete models. These tables provide critical values for a wide range of parameter settings, offering a valuable resource for researchers. Also, we present a novel application of these tests in the field of finance, specifically analyzing ‘price runs’ in financial indices. The study of price runs, a variable that reflects uninterrupted trends in market prices, has been relatively underexplored, making this a significant contribution to both statistical methodology and financial research. Moreover, we have made the programs developed for conducting these goodness-of-fit tests publicly available, ensuring that the academic community can easily replicate and extend our results.

5 - Authors used MLE to find their estimator; can any other estimation method be used? Can they add more studies on this estimator, such as unbiased property?

We use MLE because it is preferred over other estimation methods due to its consistency, asymptotic efficiency, asymptotic normality, and flexibility. Although other methods may be useful in specific contexts, MLE generally offers the best statistical properties under standard conditions.

We would like to clarify that our focus is on the presented goodness-of-fit tests rather than on the estimation methods themselves. Our tests can be applied regardless of whether the parameters are estimated using MLE, the Method of Moments, or any other estimation method biased or not. The performance and conclusions of the tests remain consistent, as they are designed to assess the goodness of fit between the observed data and the theoretical geometric distribution model.

6 - What symbol does infinity refer to in Tables 1 and 2? What size do authors use to write this symbol?

The infinity symbol (refers to the theoretically calculated values based on the asymptotic distribution, which is derived when the sample size approaches infinity. 

Regarding the size of the infinity symbol in Tables 3 and 4, it is set to 10pt. LaTeX uses the default font size, which is 10pt.

We have introduced the following text in our paper to explain this in detail:

“In Tables 3 and 4, the notation of infinity ( \\infty ) refers to the theoretically calculated values based on the asymptotic distribution, which represents the distribution of the test statistics W^2 and A^2 when the sample size $n$ approaches infinity. These asymptotic values provide a theoretical benchmark derived from the limiting behavior of the statistics as the number of observations grows large.”

7 - Can authors add another example from another field?

We appreciate your request for additional revisions, we understand the importance of maintaining high standards in the review process. However, we would like to kindly bring to your attention some challenges we are currently facing: During previous rounds of review, the process was delayed, as new referees were selected each time we submitted revisions. This caused some frustration among the authors, particularly as one of the authors was mistakenly invited to review their own paper three times. This led to two of our co-authors requesting to withdraw the manuscript, although we were able to convince them otherwise, which also required additional time.

Given these circumstances, we respectfully ask if it would be possible to move forward with the current version of the manuscript.

Thank you very much for your understanding and support. We remain fully committed to delivering a high-quality contribution to the journal and appreciate all the effort invested in the review process.

8 - The English language should be revised.

English has been revised and errors corrected.

9 - All equations must be revised to see if they need a comma, a point, or a number at the end of them.

All equations have been revised, and those requiring numbering have been numbered. Additionally, other punctuation and minor typos have been corrected.

Once again, we sincerely thank you for your time and thoughtful attention in reviewing our manuscript.

On behalf of all authors.

Dr. Alejandro Raúl Hernández Montoya

Instituto de Investigaciones en IA

Universidad Veracruzana

Xalapa, Veracruz. México

alhernandez@uv.mx

www.uv.mx/personal/alhernandez

---

## [Editor Report · Decision Letter 3]

2 Dec 2024

Anderson-Darling and Watson tests for the geometric distribution with estimated probability of success.

PONE-D-23-22495R3

Dear Dr. Hernández-Montoya,

We’re pleased to inform you that your manuscript has been judged scientifically suitable for publication and will be formally accepted for publication once it meets all outstanding technical requirements.

Kind regards,

Ahmed M. Gemeay

Academic Editor

PLOS ONE
---

## [Editor Report · Acceptance letter]

15 Dec 2024

PONE-D-23-22495R3 

PLOS ONE

Dear Dr. Hernández-Montoya, 

I'm pleased to inform you that your manuscript has been deemed suitable for publication in PLOS ONE. Congratulations! Your manuscript is now being handed over to our production team.

Kind regards, 

on behalf of

Dr. Ahmed M. Gemeay 

Academic Editor

PLOS ONE